# DataProphet: Demystifying Supervision Data Generalization in Multimodal LLMs

**Xuan Qi**$^{\gamma\tau*}$  **Luxi He**$^{\rho}$  **Dan Roth**$^{\gamma}$  **Xingyu Fu**$^{\gamma\rho}$

$^{\gamma}$University of Pennsylvania  $^{\tau}$Tsinghua University  $^{\rho}$Princeton University

🌐 Website: https://dataprophet26.github.io/  ⓞ Code  🤗 Dataset

## Abstract

Conventional wisdom in selecting supervision data for multimodal large language models (MLLMs) is to prioritize datasets that are intuitively similar to the target task (*e.g.* text-rich *v.s.* vision-centric). However, it remains unclear how reliably such similarity translates into improved performance on the test benchmarks. In this paper, we take the first step to study the problem in MLLMs: can we predict a training data's influence on a target benchmark *even before* any training takes place? To answer this question, we first conduct an in-depth analysis using 14 vision-language datasets covering 7 diverse tasks. Our analysis shows that intuitive task similarity is unreliable in predicting task generalizability, and that transfer depends on the specific dataset rather than the broader task category. We propose DataProphet, a training-free, simple yet effective metric based on multimodal perplexity, similarity, and data diversity.Our experiments demonstrate that the influence rankings for different supervision datasets derived from DataProphet is strongly-correlated with rankings based on the actual performance increase after training, with a Kendall's $\tau$ correlation coefficient of 86.0%. Moreover, we show that DataProphet can help select better supervision data, achieving up to 6.9% improvement in average over uniform selection, 1.4% over SoTA training-based baseline, and 0.2% higher than oracle experiment performance-based selection. Our code and data will be released.

## 1 Introduction

Training data is one of the most important deciding factors for Multimodal Large Language Model (MLLM) performance (Li et al., 2023; Albalak et al., 2024; Sachdeva et al., 2024; Bai et al., 2023; Wang et al., 2024b; Bai et al., 2025; Zhu et al., 2025; OpenAI, 2023; Team et al., 2023). Given the vast amount of multimodal supervision datasets, how should we effectively utilize them towards certain training targets? Some prior work has explored using high-quality data and similar tasks' data during MLLM supervised fine-tuning stage (Xia et al., 2024; Wu et al., 2025). However, these methods still require some training and mainly improve supervision performance on target tasks by removing irrelevant, redundant, or low-quality data from the training mixture. In this work, we further demystify how training data affects a test benchmark's performance. We pose the following research question in an MLLM setting:

> Given a training dataset $D_i$, can we predict the influence (measured by relative performance change) on a target benchmark $T_j$ *even before* any training takes place?

To answer this question, we first conduct a comprehensive experiment utilizing 14 vision-language fine-tuning datasets covering 7 diverse tasks (2 for each task): OCR, chart understanding, spatial reasoning, counting, knowledge-based QA, document understanding, and map understanding. We

---

*This work was completed during Xuan Qi's summer research internship at the University of Pennsylvania. Correspond to <Xuan Qi: qi-x22@mails.tsinghua.edu.cn>, <Xingyu Fu: xingyufu@princeton.edu>

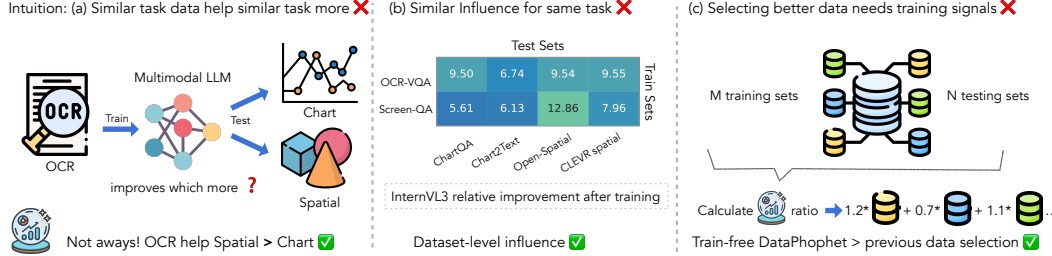

Figure 1: **Three major takeaways in DATAPROPHET**: (a) Surprisingly, human intuition about similarity between training data and test data turns out to be a poor indicator of model performance influence. In contrast, our designed DATAPROPHET metric more reliably predicts the influence of training data on test benchmarks; (b) the impact of multimodal supervision is decided by specific individual datasets, rather than by broad task categories: datasets from the same task category do not necessarily help each other the most, and do not share similar influence on the same target benchmark. Here, OCR-VQA (Mishra et al., 2019) and Screen-QA (Hsiao et al., 2022) are both OCR data, ChartQA (Masry et al., 2022) and Chart2Text (Kantharaj et al., 2022) are both chart question answering benchmarks, and Open-Spatial Cheng et al. (2024) and CLEVR spatial (Johnson et al., 2017) are both spatial reasoning benchmarks; (c) DATAPROPHET provides an effective approach for training-free data selection under fixed compute budgets (*i.e.* fixed number of total training samples). We compute influence of each supervision dataset based on a combined set of testing benchmarks and select data according to the influence ratio, yielding consistent improvements across 14 tasks, with average gains of +3.4% and +6.9% under real and synthetic data settings.

surprisingly find that different training datasets' influence on a target benchmark may contradict human intuition. For example, humans may intuitively assume that training the model on OCR task data will help its performance on chart tasks more than spatial reasoning tasks, since OCR and chart both require extracting text and numbers in an image. However, experiment results as in Figure 1(a) and (b) indicate the reverse: training on OCR improves spatial reasoning tasks more than chart tasks. This counter-intuitive finding suggests that model's generalization across different tasks depends on more than surface-level similarity. We also find that performance improvement are not decided by task category but are dependent on individual datasets: datasets from the same task category do not necessarily help each other the most, and do not share same influence on the same target benchmark.

Based on our analysis, we extract factors that *truly matter* for predicting such influence. We introduce DATAPROPHET, an extremely simple yet effective, training-free metric which uses text and visual similarity, multimodal perplexity, and data diversity to predict the data influence (Equation (3)). We evaluate the ranking of 14 training datasets' influence compared to ranking using real training effect, measured by the benchmark's relative improvement over baseline. Our experiments show strong correlation between the two rankings (86.0% Kendall's $\tau$), demonstrating the usefulness of DATAPROPHET in predicting data influence rankings in scale. Notably, the most important factors for prediction are multimodal perplexity and visual similarity, increasing 37.3% and 23.5% of Kendall's $\tau$ respectively.

In addition to predicting source data usefulness, we also show that given target benchmarks, DATAPROPHET can be applied to curate good instruction-tuning data under fixed compute and data budget. We conduct experiments under two different settings: selecting instruction-tuning data from (1) real data pool and (2) synthetic data pool. For real data, DATAPROPHET performs 3.4% better than uniform selection and is 1.4% better than ICONS (Wu et al., 2025), a SoTA selection requiring training. We observe even larger performance gain in synthetic data selection, reaching almost 6.9% performance increase compared to the uniform selection baseline and 1.2% improvement over ICONS. These show promise for DATAPROPHET as a light-weight yet effective measure for choosing supervision data for MLLMs.

Overall, this work performs a deep analysis of different source datasets' influence on target benchmarks' performance, with counter-intuitive findings. Our analysis motivates DATAPROPHET, a simple training-free yet effective metric to predict the usefulness of different training datasets on the target. We further demonstrate the promising effect of DATAPROPHET to guide MLLM instruction-tuning data selection that outperforms SoTA training-based selection.

## 2 DATA INFLUENCE ANALYSIS

In this section, we demonstrate how we quantitatively analyze data influence across multiple visual question answering (VQA) tasks. Specifically, we first select a base Multimodal Large Language Model (MLLM): InternVL3 (Zhu et al., 2025), and 14 diverse VQA datasets that comes with both train (source) and test (target) sets. Then, we conduct supervised fine-tuning on our base model with each source dataset individually, and evaluate relative performance gains on all target sets. The goal is to demystify data influence from real experiment results. We will expand this section by introducing the experiment setup, followed with results and observations.

### 2.1 ANALYSIS SETUPS

**Benchmark and Task Selection.** We try to include as diverse benchmarks as possible and mainly select three types of tasks: text-rich tasks such as OCR, chart, and document; general VQA tasks; and vision-centric tasks such as spatial reasoning, counting, and map understanding. This ends in 14 vision–language datasets spanning seven task families (2 datasets for each task), as detailed below:

- **OCR:** OCR-VQA (∼1002K QA pairs) (Mishra et al., 2019); ScreenQA (∼86K QA pairs) (Hsiao et al., 2022). This task requires extracting specific text from text-rich images and reasoning about its context.
- **Chart understanding:** ChartQA (∼33K QA pairs) (Masry et al., 2022); Chart2Text (∼44K QA pairs) (Kantharaj et al., 2022). Models must interpret chart data, identify trends, and answer questions or generate textual descriptions of charts.
- **Document understanding:** DocVQA (∼50K QA pairs) (Mathew et al., 2021); Sujet-Finance-QA (∼107K QA pairs) (Sujet AI, 2024). This task requires extracting relevant information from structured and unstructured text in documents to answer questions.
- **General VQA:** A-OKVQA (∼24.9K QA pairs) (Schwenk et al., 2022); VC-GVQA extracted from Visual-CoT (∼438K QA pairs) (Shao et al., 2024). This task focuses on general domain question-answering over natural images.
- **Spatial reasoning:** Open-Spatial (∼8.7M QA pairs) (Cheng et al., 2024); CLEVR-Relation (∼160K QA pairs) (Johnson et al., 2017). This task tests model's ability to understand spatial relationships between objects, and perform relational reasoning.
- **Counting:** CLEVR-Counting (∼160K QA pairs) (Johnson et al., 2017); TallyQA (∼287K QA pairs) (Acharya et al., 2019). This task requires accurately counting objects in images or identifying quantities based on visual cues.
- **Map reasoning:** GeomVerse (∼23K QA pairs) (Kazemi et al., 2023); MapQA (∼800K QA pairs) (Chang et al., 2022). This task focuses on understanding maps, spatial navigation, and reasoning about geographic or structural layouts.

Unless otherwise noted, we use the officially released splits or collections. For CLEVR-based subsets (CLEVR-Relation and CLEVR-Counting), we follow the question-family filters described in Johnson et al. (2017) and train/evaluate on their corresponding target sets.

**Experiment Setup.** For each benchmark $\mathcal{D}$, we uniformly sample 20k examples from their training splits for supervised fine-tuning, and 1k from test splits for evaluation. We denote the training (source) data as $s$, and testing (target) data as $t$. All experiments are conducted with a fixed compute setting – same number of training data (20K), same base model, and same hyper-parameter choices (optimizer, scheduler, epoch count, and batch size). The base model we use here is **InternVL3-2B**[1] (Zhu et al., 2025). The implementation details can be found in subsection A.2.

**Relative Performance Gain.** Here, we introduce how we measure the data influence in real experiment results. Let $A_t$ denote the base model's performance score on test set $t$ (e.g., accuracy), and $A_t^s$ the score after fine-tuning on source dataset $s$. We quantify the source-to-target influence via *relative improvement* $\Delta_{s \to t} = \frac{A_t^s - A_t}{A_t}$. Similarly, define $\Delta_{t \to s} = \frac{A_s^t - A_s}{A_s}$, where $A_s$ is the base score on $s$ and $A_s^t$ is the score on $s$ after fine-tuning on $t$.

---

[1]https://huggingface.co/OpenGVLab/InternVL3-2B

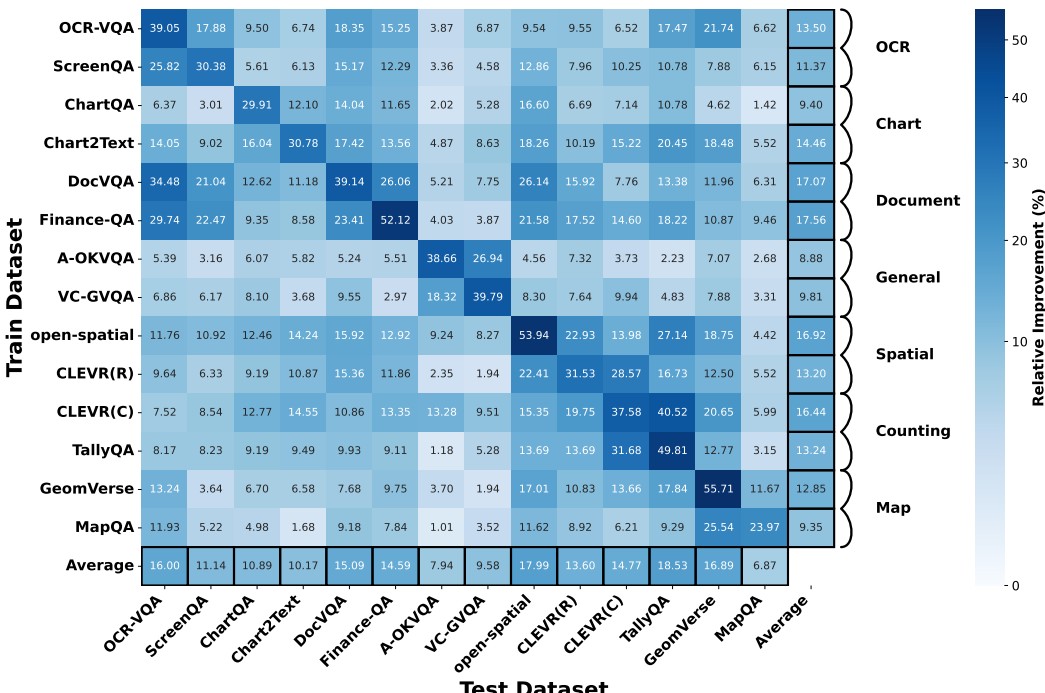

Figure 2: **Data influence analysis under fixed compute.** We conduct supervised fine-tuning on base model (InternVL 3 (Zhu et al., 2025)) with each training (source) dataset individually, and evaluate *relative performance gains* on all test (target) sets. See details in Section 2. We highlight several observations: (1) Data influence is not symmetric. (2) Data sources from the same task type do not always influence each other most. E.g., OCR-VQA improves ScreenQA (OCR task) by 17.88% vs. GeomVerse (map understanding task) by 21.74% (3) Data influence is not decided by task type, but by the individual dataset. E.g., text-rich tasks (ScreenQA) can influence vision-centric ones (Open-Spatial with gains of 12.86%) more than text-rich ones (ChartQA with gains of 5.61%).

## 2.2 ANALYSIS RESULTS

We finetune our base model with each source dataset individually, and evaluate *relative performance gains* on all target sets, to demystify data influence through controlled real experiment results. We report the full experiment results as in Figure 2. We highlight the below observations:

**Data influence is not symmetric**. The diagonal gains are still the largest, which align with our intuition that data coming from the same distribution is still the most influential. However, we find that the data influence is not symmetric, i.e. $\Delta_{s \to t} \neq \Delta_{t \to s}$. For instance, the influence of Open-Spatial on DocVQA (train on Open-Spatial and test on DocVQA) is 15.92% while the reverse is high as 26.14%.

**Data sources from the same task type do not always influence each other most**. Data influence is not decided by task type, but by the individual dataset. For instance, after training on OCR-VQA data, the relative performance gain achieved on ScreenQA (which is also OCR) is 17.88%, lower than that achieved on GeomVerse (which is on map understanding) 21.74%.

**Text-rich tasks can influence vision-centric ones more than text-rich ones**. As demonstrated in Figures 1 and 2, after training on OCR data (ScreenQA), the relative performance gain model achieves on chart tasks – 5.61% for ChartQA and 6.13% for Chart2Text – is lower than that on spatial reasoning tasks (12.86% for Open-Spatial and 7.96% for CLEVR-Relation).

## 3 DATAPROPHET: DEMYSTIFY DATA INFLUENCE

Section 2 shows that intuitive similarity is unreliable in predicting task generalizability. In this section, we investigate what the true underlying factors for predicting data influence could be. Our

goal is to identify a reliable data influence predictor based on these factors. We begin with defining a evaluation for such dataset influence prediction metrics (§3.1). We then introduce details of our metric (§3.2). Our metric is motivated by some heuristic functions that people have studied in Large Language Models (LLMs) research, such as data perplexity, task difficulty, and answer length (Marion et al., 2023; Xie et al., 2023b; Wang et al., 2024a), and we try to study their impact in multimodal settings. Building on these heuristics, we identify a training-free metric for MLLMs that reliably predicts the actual data influence results, not only for $\Delta_{s \to t}$ but also for $\Delta_{t \to s}$. We conclude the section with results of our metric and analysis including different components' effect, and ineffective but heuristic ones that we explored(§3.3).

## 3.1 PREDICTOR EVALUATION SETUP

Considering the *asymmetric nature* of data influence discussed previously, a good data influence predictor should be one that (1) produces valuable insights on how different datasets compare in contributing to some target task, and (2) offers understanding on how one dataset could affect various downstream targets differently. The ground-truth for these can be approximated by the actual relative performance gain after finetuning and evaluation, following the setup in Section 2.

**Two-Way Evaluation protocol.** For each pair of (source dataset $s$, target data $t$), we compute the influence prediction score between them using the data influence predictor. Since we have 14 datasets, we get 256 (14×14) scores in total. Given one target $t$ dataset, we rank all 14 source datasets using the influence prediction score (estimate of training data's helpfulness), and compare that with the real ranking using the relative improvement scores as in Figure 2 (true helpfulness). To quantify the correlation between two rankings, we deploy Kendall's $\tau$. We denote the correlation as $\tau_t$ for measuring $\Delta_{s \to t}$, and the average across all target datasets is $\overline{\tau}_{\text{Tgt}}$.

Conversely, given one source $s$ dataset, we rank all target datasets based on the influence score between them. Then we similarly compare it with the real rankings as in Figure 2. We denote the Kendall's $\tau$ correlation as $\tau_s$ in this setting for measuring $\Delta_{t \to s}$, and the average across all source datasets is $\overline{\tau}_{\text{Src}}$. We use

$$\overline{\tau} = \frac{\overline{\tau}_{\text{Tgt}} + \overline{\tau}_{\text{Src}}}{2}$$

as the final evaluation metric for influence predictor. Here, the two-way protocol allows for a more comprehensive evaluation of the data influence predictor itself. However, when using the influence predictor to select better training data as in Section 4, we only need to select by $\overline{\tau}_{\text{Tgt}}$.

## 3.2 THE DATAPROPHET METRIC

There are three major components in our prediction metric, building on previous work in LLM-setting and incorporating MLLM-specific considerations:

**Multimodal perplexity.** Previous studies (Marion et al., 2023; Thrush et al., 2024a) have explored to use perplexity as data quality estimator in language model pretraining, ranking and pruning pre-training corpora accordingly to achieve better model performance. Especially Thrush et al. (2024a) points out the perplexity-benchmark correlations in LLMs. Inspired by them, we examine if multimodal perplexity is the best factor for this predictor. We hypothesize that intrinsically more challenging source data can provide greater value for enhancing model capability. We define multimodal perplexity as below:
Let $A = (a_1, \ldots, a_T)$ be the answer tokens, and $p_\theta$ the base MLLM. We define multimodal perplexity on a dataset $\mathcal{D}$ as

$$\text{PPL}(\mathcal{D}) = \exp\left(-\mathbb{E}_{(I,Q,A)\sim\mathcal{D}}\,\mathbb{E}_{t\sim[T]}\,\log p_\theta\big(a_t \,|\, I,\, \tau_Q(Q),\, a_{<t}\big)\right), \tag{1}$$

We evaluate the data influence rankings using only multi-modal perplexity as predictor, and find that the resulting $\overline{\tau}_{\text{Tgt}}$ is 0.274, which is not high enough, indicating that additional components are needed for a better predictor.

**Cross-dataset Similarity.** If the source dataset contains images and text similar to the target in the model's embedding space, then heuristically it should more closely resemble learning in the target domain's distribution. We measure alignment along *questions*, *answers*, and *images* using the output

| | OCR | Chart | Spatial | Counting | General VQA | Document | Math | Avg |
|---|---|---|---|---|---|---|---|---|
| $\tau_{\text{Tgt}}$ | OCR-VQA: 0.912
ScreenQA: 0.869 | ChartQA: 0.911
Chart2Text: 0.869 | Open-Spatial: 0.736
CLEVR(R): 0.824 | CLEVR(C): 0.846
TallyQA: 0.822 | A-OKVQA: 0.868
VC-GVQA: 0.842 | DocVQA: 0.890
FinanceQA: 0.890 | GeomVerse: 0.911
MapQA: 0.933 | 0.863 |
| $\tau_{\text{Src}}$ | OCR-VQA: 0.846
ScreenQA: 0.911 | ChartQA: 0.869
Chart2Text: 0.822 | Open-Spatial: 0.822
CLEVR(R): 0.846 | CLEVR(C): 0.869
TallyQA: 0.846 | A-OKVQA: 0.846
VC-GVQA: 0.822 | DocVQA: 0.869
FinanceQA: 0.911 | GeomVerse: 0.846
MapQA: 0.869 | 0.857 |

Table 1: For each target dataset under the 7 task categories, we compute Kendall's $\tau$ between the ranking of source datasets by observed relative improvements $\{\Delta_{s\rightarrow t}\}_s$ and the ranking induced by the training-free DATAPROPHET scores $\{\mathcal{M}(s\rightarrow t)\}_s$. The high correlation coefficients show the effectiveness of DATAPROPHET in predicting the contribution ranking of different source datasets.

of the (frozen) encoders of the base MLLM, and calculate the similarity for each of them separately. For each field of question, answer and image, we encode samples with appropriate templates, then do the $\ell_2$-normalization for the embeddings, and take the *expected cosine similarity* between independently drawn items from $\mathcal{D}$ and $\mathcal{T}$. We denote these expectations by *QSim*, *ASim*, and *ISim*. This factor rewards sources whose question and answer formats and visual layouts resemble the target. Interesting, we first tried to encode question and answer together as *QASim*, but it showed worse than separate version by 6% on $\overline{\tau}$ score. We evaluate the data influence rankings using only multi-modal perplexity times similarity as predictor, and find that the resulting $\overline{\tau}_{\text{Tgt}}$ is 0.658, still not satisfactory.

**Source dataset question diversity.** Heuristically, if a source dataset contains a diverse set of questions, fine-tuning on it may encourage the model to acquire more generalizable skills, as previous papers have pointed out in LLMs (Wang et al., 2024a). To quantify this diversity, we construct an embedding representation $\mathbf{z}_u$ for each source data's example question $u$. We then cluster the embeddings into $K$ groups by certain clustering algorithms such as K-means, and compute the silhouette coefficient (Rousseeuw, 1987) given by

$$\text{Sil} = \mathbb{E}_{u\sim\mathcal{U}}\left[\frac{b(u) - a(u)}{\max\{a(u),\, b(u)\}}\right], \tag{2}$$

where $a(u)$ is the mean intra-cluster distance and $b(u)$ is the minimum mean distance from $u$ to any other cluster (both measured in the embedding space). We also quantify cluster *balance* via the normalized entropy $\text{H} = -(\log K)^{-1}\sum_{k=1}^{K}\pi_k\log\pi_k$ with $\pi_k$ the empirical cluster proportion. The final diversity is thus $\text{Sil}+\text{H}$, which larger value indicating greater coverage and more balanced, well-separated clusters. In experiments, we construct $\mathbf{z}_u$ by averaging field embeddings and apply $k$-means algorithm for clustering with $K = 10$.

**DATAPROPHET Metric.** Combining the three components above, we introduce our *training-free*, simple yet effective, data influence predictor DATAPROPHET

$$\mathcal{M}(s\rightarrow t) = \frac{\text{QSim} \cdot \text{ASim} \cdot \text{ISim} \cdot \text{PPL}(s) \cdot (\text{Sil} + \text{H})}{\text{PPL}(t)}, \tag{3}$$

where s is the training (source) data and t is the testing (target) data. DATAPROPHET implies that successful transfer requires simultaneous alignment in text, vision, difficulty to base model, and question coverage. This thus motivates the product form in our metric, which down-weights the score when any single factor is weak. We use finite-sample estimates for all expectations; when subsampling is required, we select uniformly from the full set.

### 3.3 PREDICTOR RESULTS AND ANALYSIS

**DATAPROPHET is simple but effective.** While heuristically based and training-free, our results in Table 1 suggests that DATAPROPHET can successfully predict the data reflectance as in real training results. The $\overline{\tau}_{\text{tgt}}$ is consistently high across all target datasets and task types, with an average value of 0.863. And the $\overline{\tau}_{\text{src}}$ also reaches 0.857 across all source datasets, indicating the effectiveness of our metric.

**Importance of individual components.** We further investigate the contribution of each component to our metric. Removing each factor in Equation (3) degrades rank accuracy (Table 2). The most pronounced drop occurs when *multimodal perplexity* is removed ($0.863 \rightarrow 0.491$), highlighting the

| Variant | Kendall's $\tau$ (Avg) |
|---|---|
| Full DATAPROPHET | **0.860** |
| w/o Answer Similarity | 0.810 |
| w/o Question Similarity | 0.778 |
| w/o Diversity (Silhouette & Entropy) | 0.659 |
| w/o Image Similarity | 0.625 |
| w/o Perplexity | 0.487 |

Table 2: **Ablation of DATAPROPHET components** Each row indicates removing one factor from Equation equation 3 and recompute Kendall's $\tau$. Different components have varying effects on the correlation coefficient.

value of modeling source headroom and target difficulty. *Image similarity* (0.627) and *diversity* (0.658) are also crucial components, suggesting that visual alignment and coverage are essential for transfer. Textual similarities from questions and answers are useful, but their contributions are more marginal compared to the other stronger signals.

**Additional trials.** We conduct multiple additional heuristic-based trials but they all fail to show improvement: question difficulty, model's familiarity with the image, model's familiarity with the question, and answer length. For example, after incorporating answer length, the $\bar{\tau}$ decreased by 0.15. Only the three major factors in our DATAPROPHET metric remains effective.

## 4 DATAPROPHET FOR DATA SELECTION

Besides predicting the usefulness ranking of different source datasets, DATAPROPHET can also be used to guide data selection *under a fixed compute budget*. We first introduce two selection setups for supervised instruction tuning: real-data reweighting (§4.1) and synthetic-data ranking and selection (§4.2). We then present the outcomes and analysis in §4.3. Finally, we include an additional study on RL post-training data selection (Table 4).

### 4.1 REAL-DATA REWEIGHTING

We fix the total training budget to be $N=14 \times 20K = 280K$ samples. Using the fourteen vision–language datasets as described in Section 2 as candidate sources, we determine each dataset's relative weighting in the mixture using their DATAPROPHET score. Intuitively, datasets assigned higher DATAPROPHET scores are expected to contribute more positively to target performance, and are therefore allocated a larger proportion of samples in the mixture. For more details on our reweighting algorithm, please see Appendix A.3.

Note that the relative weighting would sometimes require more than $20K$ items from a given dataset, which is beyond the size of our original 14 datasets. In such scenarios, we sample new data from the original full data source to obtain the desired amount.

For the real-data reweighting setup, we compare the following baseline methods of obtaining the training set of 280K training samples for a given target set.

**Uniform**: combine the 14 source datasets with 20K examples uniformly sampled from each.

**Oracle**: use observed relative improvements from Figure 2 to reweigh the source datasets.

**ICONS** (Wu et al., 2025): applies the gradient-based influence consensus approach to score all data in the original real data pool, then use the top 280K datapoints as the training set. This training-required method is current state-of-the-art data selection method for MLLM instruction tuning.

**D.P.**: use DATAPROPHET to reweigh the source datasets.

| Benchmarks / Sampling Methods | Real Data Reweighting | | | | Synthetic Data Selection | | |
|---|---|---|---|---|---|---|---|
| | Baseline | | Gold | Ours | Baseline | | Ours |
| | Uniform | ICONS | Oracle | D.P. | Uniform | ICONS | D.P. |
| OCR-VQA (Mishra et al., 2019) | 85.2 | 86.5 | 87.5 | 87.4 | 65.4 | 73.5 | 74.6 |
| ScreenQA (Hsiao et al., 2022) | 83.1 | 83.6 | 84.8 | 85.0 | 67.3 | 76.5 | 78.4 |
| ChartQA (Masry et al., 2022) | 83.7 | 84.8 | 84.9 | 85.6 | 68.2 | 77.2 | 78.3 |
| Chart2Text (Kantharaj et al., 2022) | 84.0 | 84.5 | 86.0 | 86.2 | 68.7 | 75.9 | 77.5 |
| Open-Spatial (Cheng et al., 2024) | 37.9 | 40.7 | 41.2 | 41.1 | 26.7 | 27.3 | 28.5 |
| CLEVR(R) (Johnson et al., 2017) | 41.5 | 42.2 | 45.0 | 45.6 | 33.8 | 35.2 | 37.6 |
| CLEVR(C) (Johnson et al., 2017) | 44.5 | 45.3 | 47.2 | 48.1 | 35.6 | 36.8 | 38.6 |
| TallyQA (Acharya et al., 2019) | 41.8 | 42.8 | 45.2 | 45.1 | 30.4 | 35.5 | 36.2 |
| A-OKVQA (Schwenk et al., 2022) | 83.7 | 84.9 | 85.8 | 86.0 | 68.2 | 74.4 | 74.8 |
| VC-GVQA (Shao et al., 2024) | 81.5 | 82.3 | 83.7 | 83.5 | 67.8 | 75.9 | 76.8 |
| DocVQA (Mathew et al., 2021) | 76.9 | 79.4 | 80.4 | 81.4 | 58.6 | 66.4 | 67.2 |
| FinanceQA (Sujet AI, 2024) | 73.8 | 74.1 | 75.1 | 75.2 | 57.2 | 64.8 | 65.9 |
| GeomVerse (Kazemi et al., 2023) | 60.1 | 61.2 | 62.4 | 63.0 | 47.1 | 53.1 | 54.4 |
| MapQA (Chang et al., 2022) | 82.9 | 83.3 | 83.5 | 84.8 | 75.2 | 78.2 | 79.8 |
| Average | 67.6 | 69.6 | 70.8 | **71.0** | 55.1 | 60.8 | **62.0** |
| Improve over Uniform | - | +2.0 | +3.2 | **+3.4** | - | +5.7 | **+6.9** |

Table 3: **Data selection results.** Keeping a fixed compute budget of 280K datapoints, we present the data selection results on 14 target benchmarks. Sampling methods: `Uniform` is the uniform sampling baseline, `ICONS` is the sota training-based baseline, and `Oracle` is the gold baseline that selects samples based on oracle real experiment performance improvement as in Figure 2. `Average` reports the mean performance across all benchmarks, and *Improve over Uniform* shows the absolute performance gain relative to uniform sampling. From the results, our DATAPROPHET-based sampling achieves the best average performance on both real and synthetic data, surprisingly reaching up to +0.2% improvement over the gold oracle selection.

## 4.2 SYNTHETIC-DATA SELECTION

Given a target dataset and a large pool of synthetic training candidates, DATAPROPHET can also be used to identify the subset of data expected to be most beneficial for improving target performance. We score each individual datapoint $syn\_d$ by a simplified version of the original metric which removes the diversity term:

$$\mathcal{M}(syn\_d \rightarrow \mathcal{T}) = \frac{\text{QSim} \cdot \text{ASim} \cdot \text{ISim} \cdot \text{PPL}(Syn\_D)}{\text{PPL}(\mathcal{T})}. \quad (4)$$

In our experiments, we construct the synthetic data pool via the following: For each of the 14 datasets, we split its 20k images into two disjoint halves of 10k images each. One half is sent to GPT-5 and the other half to Gemini 2.5 Pro. For every image, the generator is asked to produce five VQA-style question–answer pairs conditioned only on the image. Both GPT and Gemini 2.5 Pro receive the *same* prompt template (Appendix A.4). This yields $10k \times 2 \times 5 = 100k$ QA pairs per dataset and $\sim 1.4M$ pairs in total across the 14 datasets. We retain all pairs that satisfy basic validity checks (non-empty strings, short answers without special tokens).

Using this synthetic data pool, we compare the following baseline data selection methods: **Uniform**, which uniformly samples 280k data from the full synthetic data pool and **ICONS** (Wu et al., 2025), to our method **D.P.** for DATAPROPHET.

| Benchmarks / Methods | No RL | Real RL Data Selection | | Synthetic RL Data Selection | |
| --- | --- | --- | --- | --- | --- |
| | *Baseline* | *Baseline* | *Ours* | *Baseline* | *Ours* |
| | Baseline | Equal | D.P. | Random | D.P. |
| OCR-VQA (Mishra et al., 2019) | 72 | 75 | 76 | 71 | 73 |
| ScreenQA (Hsiao et al., 2022) | 66 | 68 | 69 | 68 | 67 |
| ChartQA (Masry et al., 2022) | 70 | 71 | 73 | 69 | 72 |
| Chart2Text (Kantharaj et al., 2022) | 68 | 70 | 72 | 70 | 69 |
| Open-Spatial (Cheng et al., 2024) | 35 | 39 | 41 | 36 | 37 |
| CLEVR(R) (Johnson et al., 2017) | 41 | 44 | 46 | 42 | 42 |
| CLEVR(C) (Johnson et al., 2017) | 49 | 52 | 52 | 50 | 52 |
| TallyQA (Acharya et al., 2019) | 31 | 34 | 33 | 33 | 35 |
| A-OKVQA (Schwenk et al., 2022) | 64 | 68 | 69 | 62 | 65 |
| VC-GVQA (Shao et al., 2024) | 65 | 67 | 69 | 64 | 65 |
| DocVQA (Mathew et al., 2021) | 59 | 62 | 60 | 62 | 63 |
| FinanceQA (Sujet AI, 2024) | 54 | 55 | 56 | 55 | 54 |
| GeomVerse (Kazemi et al., 2023) | 42 | 44 | 47 | 43 | 45 |
| MapQA (Chang et al., 2022) | 64 | 67 | 66 | 64 | 68 |
| *Average* | 55.7 | 58.3 | **59.5** | 56.4 | **57.7** |
| *Improve* over Baseline | - | +2.6 | **+3.8** | +0.7 | **+2.0** |

Table 4: **RL post-training data selection results.** We fix a prompt budget of $300 \times 14$ and evaluate on the same 14-benchmark test suite. For real RL data, we compare equal allocation (Equal) to DATAPROPHET-guided allocation (D.P.). For synthetic RL data, we compare random sampling (Random) to DATAPROPHET-guided selection. *Average* reports the mean score across benchmarks, and *Improve over Baseline* shows the absolute gain relative to the no-RL baseline.

## 4.3 RESULTS AND ANALYSIS

**DATAPROPHET provides effective training-free guidance for data selection.** For each of the 14 target benchmark datasets, we conduct real-data reweighting and synthetic-data selection using the methods above. Table 3 summarizes the results for both selection paradigms. Overall, DATAPROPHET-guided selection achieves the best macro-average performance across tasks in both the real and synthetic setups, yielding 3.4% and 6.9% improvements, respectively. Notably, reweighting guided by DATAPROPHET matches or exceeds the **Oracle** reweighting constructed from observed performance gains. Moreover, our training-free metric outperforms the training-based selector ICONS, demonstrating that DATAPROPHET is both effective and computationally efficient for data selection.

**Does Gemini 2.5 Pro or GPT 5 provide higher-quality supervision data?** Interestingly, it turns out that Gemini produces higher-quality synthetic data than GPT. In the synthetic data setup, about 38% of selected items come from GPT-5 and 62% from Gemini 2.5 Pro, indicating that Gemini contributes a larger share of items that score highly under the instance-level DATAPROPHET metric. Manual inspection suggests that these items are more challenging, reflected by higher per-token perplexity, and better aligned with the target's distributions in layout, entities, and answer forms.

**DATAPROPHET also improves RL post-training data selection.** We further evaluate whether DATAPROPHET can guide data selection for RL post-training. We perform GRPO training using the `verl` implementation on Qwen2.5VL-3B-Instruct with default hyperparameters, and fix the RL prompt budget to $300 \times 14$. We compare equal allocation across the 14 datasets to a DATAPROPHET-guided allocation for real RL data, and random sampling to DATAPROPHET-guided selection for synthetic RL data. As shown in Table 4, DATAPROPHET improves the average score from 0.583 to 0.595 for real RL data and from 0.564 to 0.577 for synthetic RL data on the 14-benchmark test suite.

## 5 RELATED WORKS

**Data Mixtures for LMs**   Data Mixture has long been an interesting problem in the research community. Xie et al. (2023a) propose DoReMi, which trains a small proxy model with GroupDRO to learn domain weights and then resamples the corpus for full-scale training. A complementary line fits predictive mixture laws (Ye et al., 2025; Ge et al., 2025; Kang et al., 2025; Li et al., 2025). Beyond domain-level mixtures, example-level selection has been studied for both pretraining and instruction tuning, such as DSIR (Xie et al., 2023b) and LESS (Xia et al., 2024). Simple perplexity-based pruning and perplexity correlations offer strong, training-free selectors at pretraining scale (Marion et al., 2023; Thrush et al., 2024b; Qi et al., 2025). Our work differs in being training-free: instead of proxy runs, gradients, or online adaptation, we propose a simple metric that scores training sets for multi-task utility before any finetuning.

**Data Mixtures for Multimodal LMs**   In instruction tuning of VLMs, ICONS aggregates first-order influence estimates across tasks to identify broadly useful examples, attaining near-parity with full-data training using compact subsets (Wu et al., 2025). Related selection frameworks estimate task- and instance-level value using gradient features (Liu et al., 2024) or score with the model itself to filter hard/diverse instructions (Chen et al., 2024). On the curation side, DATACOMP standardizes large-scale image–text pretraining as a dataset filtering problem, enabling fair comparison across selection strategies and scales (Gadre et al., 2023). At the post-training/RL stage, MoDoMoDo formulates multimodal RLVR as a mixture-optimization problem, learning policies that improve out-of-distribution generalization (Liang et al., 2025). In contrast to these training-dependent paradigms (proxy sweeps, gradient stores, or RL rollouts), we target a training-free, pre-finetuning decision rule that is compute-light and directly applicable to both real and synthetic multimodal data.

**Domain and dataset generalization for LMs**   In the LM context, existing work explores the generalizability of SFT or RL training from one domain to another. Reinforcement post-training (RPT) can generalize beyond the fine-tuned domain under certain conditions (Hu et al., 2025); comparative studies suggest SFT tends to memorize while RL-style post-training generalizes more robustly (Chu et al., 2025); and math-specialized training shows mixed transfer unless optimized via RL rather than SFT (Huan et al., 2025). However, these works do not offer metrics for analyzing the contribution of different datasets to improving performance on the target domain. Our work seeks to both deepen understanding and offer prescriptive guidance.

## 6 CONCLUSIONS

In this paper, we investigate whether one can predict before any training how much a given source dataset will benefit a target dataset in MLLMs. Across seven task families, we find that intuitive task similarity is an unreliable guide and that transfer is dataset-specific rather than task-specific. To address this, we introduce DATAPROPHET, a simple, training-free and interpretable metric that integrates cross-modal similarity, multimodal perplexity, and source dataset diversity. DATAPROPHET closely aligns with realized training outcomes when ranking data influence, and further serves as a useful guidance for both real and synthetic data selection. We believe DATAPROPHET provides a principled signal for data curation, enabling more effective MLLM training.

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

# A APPENDIX

## A.1 THE USE OF LLMs

LLMs did not play an important role in this paper's research ideation or writing to the extent that they should be regarded as a contributor. In the experiments, LLMs are the main experimental object.

## A.2 IMPLEMENTATION DETAILS FOR SECTION 2

All runs fine-tune the base model for one epoch using full-parameter supervised fine-tuning (Full SFT) implemented in LLaMA-Factory (Zheng et al., 2024). The learning rate is set to be $1e-5$. All evaluations follow the benchmarks' task-appropriate metrics.

---

**Algorithm 1** Data Reweighting based on the metric $\mathcal{M}$

---

**Require:** target dataset $\mathcal{D}_t$; source datasets $\{\mathcal{D}_s^{(i)}\}_{i=1}^M$; total budget $N$
**Ensure:** training set $S$ for SFT
1: **for** $i \leftarrow 1$ **to** $M$ **do**
2:     $m_i \leftarrow \mathcal{M}\big(\mathcal{D}_s^{(i)} \rightarrow \mathcal{D}_t\big)$                    ▷ compute metric between source $i$ and target
3: **end for**
4: $Z \leftarrow \sum_{j=1}^M m_j$
5: **for** $i \leftarrow 1$ **to** $M$ **do**
6:     $N_i \leftarrow \mathrm{round}\big(N \cdot \frac{m_i}{Z}\big)$
7:     $S_i \leftarrow \textsc{SampleUniform}\big(\mathcal{D}_s^{(i)}, N_i\big)$
8: **end for**
9: $S \leftarrow \bigcup_{i=1}^M S_i$
10: **return** $S$

---

**Algorithm 2** Metric-guided Synthetic Data Selection

---

**Require:** target dataset $\mathcal{D}_t$; synthetic pool $\mathcal{S}$; either Top-$K$ or threshold $\tau$
**Ensure:** selected synthetic subset $\mathcal{S}_{\text{sel}}$
1: **for each** $u \in \mathcal{S}$ **do**
2:     $\mathrm{QSim}(u, \mathcal{D}_t)$, $\mathrm{ASim}(u, \mathcal{D}_t)$, $\mathrm{ISim}(u, \mathcal{D}_t)$
3:     $\mathrm{PPL}_{\text{train}}(\{u\})$, $\mathrm{PPL}_{\text{test}}(\mathcal{D}_t)$
4:     $\tilde{m}(u) \leftarrow \mathrm{QSim} \cdot \mathrm{ASim} \cdot \mathrm{ISim} \cdot \mathrm{PPL}_{\text{train}}(\{u\}) \,/\, \mathrm{PPL}_{\text{test}}(\mathcal{D}_t)$
5: **end for**
6: **if** $\textsc{UseTopK}$ **then**
7:     $\mathcal{S}_{\text{sel}} \leftarrow \textsc{TopK\_By\_Value}\big(\mathcal{S}, \tilde{m}, K\big)$
8: **else**
9:     $\mathcal{S}_{\text{sel}} \leftarrow \{\, u \in \mathcal{S} : \tilde{m}(u) \geq \tau \,\}$
10: **end if**
11: **return** $\mathcal{S}_{\text{sel}}$

---

### A.3 DETAILED ALGORITHMS

### A.4 PROMPT TEMPLATES

The following template is used for both GPT-5 and Gemini 2.5 Pro. The calling code injects the image in place of `<IMAGE>`.

```
You are generating visual question answering data strictly
grounded in the given image.  Use only what is visible
in the image.  Do not rely on outside knowledge.  These
questions should be inference questions about what is in the
picture.
Image:  <IMAGE>
Task:  Create exactly five diverse VQA pairs about this
image.
Constraints:  questions must be answerable solely from the
image; answers should be short (several words or a number);
avoid ambiguous or subjective wording; avoid near-duplicate
questions; avoid requiring reading tiny illegible text; if
no text is legible, do not ask OCR questions.
Output format:  return a single JSON array of five objects.
Each object has the following fields
    {-} "question":  one string
    {-} "answer":  one string (lowercase for each word and
numerals for numbers)
Do not include any additional text before or after the JSON.
```

