# OpenReview forum: "Demystifying Supervision Data Generalization in Multimodal LMs"
_ICLR.cc/2026/Conference — ICLR 2026 Poster_

### Official Review · Reviewer_LeY5 · 2025-10-19

**Soundness:** 2
**Presentation:** 2
**Contribution:** 2
**Rating:** 4
**Confidence:** 4

**Summary:**

This paper addresses the fundamental question of predicting training data influence on target benchmarks in multimodal large language models (MLLMs) before any training takes place. The authors conduct a comprehensive empirical analysis using 14 vision-language datasets across 7 diverse tasks and make several counter-intuitive discoveries that challenge conventional wisdom about data selection.
Key Contributions
1. Counter-intuitive Empirical Findings: The paper argues that intuitive task similarity is an unreliable predictor of cross-task generalization.
2. Dataset-level vs. Task-level Influence: The authors demonstrate that data influence operates at the individual dataset level rather than broad task categories.
3. DATAPROPHET Metric: The paper introduces a training-free, interpretable metric that combines three components:
    - Multimodal Perplexity: Measures source data difficulty relative to target
    - Cross-dataset Similarity: Captures alignment in questions, answers, and images using MLLM embeddings
    - Source Dataset Diversity: Quantifies question coverage using clustering-based silhouette coefficient and entropy

**Strengths:**

**Originality**

- The paper introduces a research question - predicting cross-dataset influence in MLLMs before any training occurs. While data selection has been extensively studied, the specific focus on training-free prediction of multimodal data influence represents a clear departure from existing gradient-based or proxy-model approaches.

- Systematic Empirical Investigation: The comprehensive 14×14 influence matrix analysis is unprecedented in scope for multimodal settings. This systematic approach to mapping cross-task transfer patterns fills an important gap in understanding MLLM behavior.

**Quality**
- The paper demonstrates experimental breadth by selecting 14 diverse vision-language datasets spanning 7 task families (OCR, chart understanding, document understanding, general VQA, spatial reasoning, counting, and map reasoning). This coverage, with 2 datasets per task type, enables robust cross-task transfer analysis. The systematic 14×14 experimental matrix (196 training-evaluation combinations) provides empirical evidence for understanding how different source datasets influence performance across various target benchmarks.
- Rigorous Experimental Design: The controlled experimental setup is well-designed with consistent hyperparameters, fixed compute budgets (20K samples each), and standardized evaluation protocols across all 14 datasets.

**Clarity**
- Well-Structured Presentation: The paper follows a logical progression from motivation → empirical analysis → method development → validation. The three-part takeaway in Figure 1 effectively communicates the main idea.

- Effective Visualization: Figure 2's heatmap clearly illustrates the nature of cross-dataset influence. The color-coding and task groupings make patterns easily interpretable.

**Significance**
The paper challenges the widespread assumption that "similar tasks help similar tasks more." The systematic influence analysis framework and the 14-dataset benchmark provide a reference for comparative studies.

**Weaknesses:**

**Single Model Architecture**: All experiments are conducted exclusively on InternVL3-2B, which severely limits the generalizability of findings. Different MLLM architectures (e.g., GPT-4V, LLaVA, BLIP families) may exhibit fundamentally different cross-task transfer patterns due to varying pretraining objectives, data distributions, and architectural choices. The authors provide no evidence that DATAPROPHET's effectiveness extends beyond this single model family, making it unclear whether the discovered "counter-intuitive" patterns are universal phenomena or model-specific artifacts.

**Ill-Defined Problem Formulation and Claims**
- Vague "Counter-Intuitive" Claims: The paper's central claims about counter-intuitive findings are problematic due to poorly defined baselines. In the Introduction, the authors state that "humans may intuitively assume that training the model on OCR task data will help its performance on chart tasks more than spatial reasoning tasks, since OCR and chart both require extracting text and numbers in an image." However, "intuitive" is not a well-defined, measurable concept. The authors provide no systematic survey of expert opinions, formal definition of intuitive similarity, or principled baseline for what constitutes "expected" transfer patterns. This makes their counter-intuitive claims essentially unfalsifiable and scientifically questionable.

- Undefined Task Categories: The paper repeatedly refers to findings being "dependent on individual datasets" rather than "task category," but "task category" itself lacks clear definition. The authors don't explain what constitutes a task boundary, how fine-grained the categorization should be, or what criteria distinguish one task category from another. This conceptual ambiguity undermines the central thesis about dataset-level vs. task-level influence. This subjective categorization makes it impossible to assess whether the reported transfer patterns reflect genuine task relationships or merely artifacts of the chosen taxonomy.

**Limited Long-term Training Analysis**: All experiments use single-epoch training, but production MLLM training typically involves multiple epochs and complex scheduling. The influence patterns observed in short training runs may not persist during extended training.

**Questions:**

**Contradictory Evidence in Task Category Claims**
There appears to be a contradiction between your claim that "datasets from the same task category do not necessarily help each other the most" (Figure 1b) and the actual results in Figure 2. Taking ChartQA as an example from the Chart understanding task family: the highest improvement comes from Chart2Text (+16.04%), which is indeed from the same task category as defined in Section 2.1. Spatial reasoning tasks like Open-Spatial (+12.46%) and CLEVR(R) (+9.19%) show lower improvements than the same-category Chart2Text. This pattern appears to support intuitive same-task-category transfer rather than contradicting it. Could you:
  - Clarify how you define "task category" boundaries for this specific analysis?
  - Explain why the ChartQA example doesn't contradict your main claims about counter-intuitive transfer patterns?
  - Provide a more systematic analysis of when same-category transfer does vs. doesn't dominate?

**Fundamental Flaws in Cross-Task Transfer Analysis**

Your Observations 2 and 3 in Section 2.2 are based on comparing relative improvements across different target datasets from the same source dataset (same row, different columns in Figure 2). However, this analytical approach suffers from critical confounding factors that invalidate your conclusions:

Core Problem: You compare relative improvements across tasks with fundamentally different baseline difficulties and improvement potentials. The tasks themselves have different intrinsic characteristics that affect their "improvability," making cross-task comparisons of relative gains meaningless.

 Specific Evidence from Your Data: For Observation 2, you claim: "after training on OCR-VQA data, the relative performance gain achieved on ScreenQA (OCR, 17.88%) is lower than that achieved on GeomVerse (map understanding,21.74%)."
   However, examining Figure 2's data reveals confounding factors:
  - GeomVerse shows consistently higher average relative gains (16.89%) compared to ScreenQA (11.14%) across ALL source datasets
  - GeomVerse's self-improvement ($\Delta_{s→s}$) is 55.71% vs ScreenQA's 30.38%
  - This suggests GeomVerse is simply more "improvable" as a benchmark, regardless of source dataset

 Questions:

  1. Confounding Control: How do you distinguish between genuine cross-task transfer effects and task-intrinsic improvability differences? Your current analysis cannot separate these factors.
  2. Baseline Normalization: Have you considered normalizing improvements by task-specific baselines or maximum achievable gains? Without such normalization, comparing raw relative
  improvements across different tasks is scientifically invalid.
  3. Alternative Explanations: How do you rule out that the observed patterns are due to:
    - Different evaluation metric sensitivities
    - Varying task saturation points
    - Benchmark design artifacts
    - Different training dynamics needed by task types
  4. Causal Claims: Your claim that transfer depends on "individual datasets" rather than "task categories" requires showing that dataset-specific factors (beyond task characteristics) drive
  the observed patterns. How do you establish this causal relationship?

  The Same Issue Applies to Observation 3: Your examples of "text-rich tasks influencing vision-centric ones more than text-rich ones" likely reflect the same confounding - vision-centric
  tasks may simply have more room for improvement rather than indicating genuine cross-modal transfer superiority.

  Impact on Paper's Validity: This analytical flaw undermines your central claims about dataset-specific vs. task-specific influence. Without proper controls for task-intrinsic factors, your
  conclusions about "counter-intuitive" transfer patterns may be statistical artifacts rather than genuine insights.

  Suggested Resolution: Re-analyze your data using improvement metrics that account for task-specific characteristics, or restrict comparisons to tasks with matched baseline difficulties and
  improvement potentials.

---

> ### Author Response · Authors · 2025-11-29
>
> Dear Reviewer LeY5,
> We sincerely thank you for the time you dedicated to reviewing our work. We thank the reviewer for the careful reading and for recognizing the originality of the problem setup and the empirical mapping of cross-dataset influence. We hope our response and added experiment results could address the concerns.
>
> **On W1 (Single model architecture and training regime)**
> Due to the constraint on compute resources, we had to limit our experiments to the base model InternVL3-2B. While solely based on this model, we want to mention that it was the state-of-the-art open-source VL model of this size – with much better performance than LLaVA and BLIP. Also, we want to emphasize that GPT-4V cannot do such experiments because it’s not open-source. While only covering InternVL3-2B, we covered a diverse 14 datasets, which is enough to prove this methods’ efficacy. We want to emphasize that many published papers all try to improve based on single specific model architecture, this is a common practice in multimodal research, e.g. [Visual CoT: Advancing Multi-Modal Language Models with a Comprehensive Dataset and Benchmark for Chain-of-Thought Reasoning from NeurIPS 2024] on LLaVa. We will leave experiments testing different MLLM structures to future work.
>
> **On W2a (Intuition Concerns)**
> Our use of the term *intuitive* is not intended to introduce a formal or measurable construct; rather, it serves as a motivational observation of a commonly held expectation in the multimodal community—namely, that tasks sharing surface-level operations (e.g., OCR and chart understanding both involving textual/numerical extraction) are often presumed to transfer positively to one another. This intuition is widely reflected in prior literature on cross-task transfer, curriculum learning, and dataset design (e.g., works such as [Which Tasks Should Be Learned Together in Multi-task Learning?] from ICML’20, [Factors of Influence for Transfer Learning across Diverse Appearance Domains and Task Types] from TPAMI, etc), which similarly rely on informal yet ubiquitous assumptions about task similarity in order to motivate empirical investigations rather than to define scientific hypotheses.
>
> **On W2b (Task category concerns)**
> Task families in our study are defined explicitly in Section 2.1 as seven coarse categories (OCR, chart understanding, document understanding, general VQA, spatial reasoning, counting, map reasoning), each with two datasets. These categories are not subjectively defined, but are objectively widely adopted in the community, used in every multimodal research paper.
>
> **On W3 (Why not longer-term training)**
> We use single-epoch SFT here instead of multi-epoch SFT training here to avoid overfitting and keep generalization capabilities, following a common practice for continual training on already well-trained MLLMs. In this work we deliberately fix the number of examples and train for one epoch on each source to isolate the relative influence of data under the same compute budget.
>
> **On Q1 (Explain ChartQA example)**
> As we mentioned above, the task categories are not subjectively defined, but are objectively widely adopted in the community, used in every multimodal research paper. There is also no contradictory claims in our paper: what we meant in the ChartQA example was – normally, humans would assume that same task data (or in-domain data) help each other more, but it’s not guaranteed to always be true- this “not always true” is the counter-intuitive finding. Our intent in Observation 2 is not to claim that same-category transfer never dominates, but that it does not always dominate, as illustrated by other entries in Figure 2—for instance, OCR-VQA improving GeomVerse (map) more than ScreenQA (OCR), or ScreenQA improving spatial reasoning tasks more than chart tasks.
>
> **On Q2 (Relative Improvement measurement concerns)**
> The reviewer claims that “The tasks themselves have different intrinsic characteristics that affect their "improvability," making cross-task comparisons of relative gains meaningless.” and we kindly disagree. Relative improvement over a baseline is the de facto standard for analyzing influence and transfer across tasks with different metrics and scales. Numerous published works—e.g. MT-CRL (NeurIPS), CrossFit (EMNLP), … etc all use relative gains as the primary evaluation metric. What we want here is an objective evaluation metric, and we do not adjust the metric by any data-specific feature.
>
> Our analysis is therefore deliberately designed to be fully observational and unbiased: all tasks are treated equally, all setups are identical within a row, and the resulting relative differences reflect genuine transfer behavior rather than any pre-imposed normalization scheme. The fact that many observed cross-task contrasts remain consistent even without any task-level adjustment suggests that these effects are not artifacts of evaluation design but intrinsic properties of dataset–task interactions.

---

### Official Review · Reviewer_SyPJ · 2025-10-27

**Soundness:** 2
**Presentation:** 3
**Contribution:** 2
**Rating:** 4
**Confidence:** 3

**Summary:**

This paper investigates the challenge of predicting a multimodal large language model’s (MLLM’s) generalization benefit from a given supervision dataset, even before training. The authors develop DataProphet, a training-free metric combining multimodal perplexity, cross-modality similarity, and dataset diversity to estimate the influence of candidate training data on downstream benchmarks. They show that traditional intuition about task similarity fails to predict which training data is most helpful, and demonstrates that its influence ranking correlates with performance gains.

**Strengths:**

- The paper presents a systematic, large-scale analysis of data generalization in MLLMs, involving 14 datasets over diverse VQA-related vision-language tasks. This scope (detailed in Section 2 and Figure 2) provides a valuable resource and a robust empirical foundation for the claims made.
- By quantitatively demonstrating (Figure 1, Figure 2) that surface-level or even task-category similarity does not reliably predict data transfer utility, the work provides a corrective to widespread "intuitive data selection" assumptions in the field.
- DataProphet leverages a mix of complexity (perplexity), alignment (similarity), and diversity factors, all easily computable without requiring costly pretraining or proxy tasks. This increases applicability in compute-limited scenarios.
- The authors use strong evaluation protocols (employing Kendall's \tau between predicted influence rankings and the ground truth) and ablation studies to show the contribution of each metric component.
- They have shown DataProphet is competitive or superior to state-of-the-art, more expensive approaches, boosting both real and synthetic data selection, and even outperforming performance-based oracle mixtures in average (Table 3).

**Weaknesses:**

- All experiments are performed using InternVL3-2B. The generality of findings to other competitive models (e.g., Gemini, GPT-4V, Qwen-VL) is not established. Since DataProphet calculations depend on the base model’s embedding space and perplexity, there is a risk of model-specific artifacts, especially when transferred to different backbone architectures.

- Critical equations such as the definition of dataset diversity (Page 6), silhouette score, and their combination with entropy lack detailed motivation for the additivity, choice of $K=10$, cluster methodology, and scaling. The rationale for normalizing perplexities when transferring across datasets is not justified (e.g., differing answer lengths, vocabulary distributions, or answer spaces could bias comparisons. There are also minor ambiguities in notation (e.g., use of $\Delta_{s \rightarrow t}$ notation on Page 3, 4) should clarify whether this strictly measures test accuracy delta or considers area under curve for training budget). Additionally, the metric for synthetic data selection drops the diversity term (Page 8) without justification or discussion of why this is valid—an omission that may have side effects for coverage in less-redundant pools.

- While the procedure is generally clear, some details are underspecified. For instance, random seeds, batch sizes, number of epochs, hardware, and exact computation of metrics like Kendall’s \tau and confidence intervals are missing. There is no assessment of the variability of DataProphet scores under resampling, nor is there a discussion of selection stability over different random seeds.

- While the DataProphet metric is intuitive and empirically effective, the paper advances no rigorous theoretical analysis of why the product formulation (Equation 3), or the selection of factors, is optimal or robust. What are the theoretical regimes or failure cases? Why use a product rather than, e.g., weighted sum, normalized version, etc.? The absence of any formal support or negative result is especially glaring given the field’s growing interest in principled data mixture theory (see, e.g., [1]).

- Minor: suggestion include some recent related works:
  - [2] related for benchmarking MLLMs under data ambiguity and generalization; should be referenced in Section 5 (Related Works) and discussed in terms of differences in experimental scope and generalization perspective.
  - [3] gives metrics for bias/generalization in model evaluation; should be mentioned when discussing selection criteria and effects of selection (Section 4, Section 5).
  - [4] is related for ongoing tuning/data mixture effects; discuss in Related Works and comparison section.

[1] Jiasheng, Ye et al. Data mixing laws: Optimizing data mixtures by predicting language modeling performance. In International Conference on Learning Representations (ICLR), 2025.

[2] Wang, Ru; Song, Selena; Ding, Liang (2025): MMA: Benchmarking Multi-Modal Large Language Model in Ambiguity Contexts

[3] Vo, An; Taesiri, Mohammad Reza; Kim, Daeyoung (2025): B-score: Detecting biases in large language models using response history

[4] Chen, Cheng; Zhu, Junchen; Luo, Xu (2024): CoIN: A Benchmark of Continual Instruction Tuning for Multimodal Large Language Models

**Questions:**

Around Table3, the discussion lacks qualitative analysis or concrete case studies of failures: When does DataProphet select harmful or unhelpful examples? How robust is the metric to data imbalance, annotation quality, or label noise in the source datasets?

And also see my embedded question in "weaknesses".

**Details Of Ethics Concerns:**

There are potential risks that DataProphet-based data selection systematically underrepresents certain modalities, domains, or linguistic/cultural groups, particularly when similarity is defined in the embedding space of an MLLM that itself may be biased.

---

> ### Author Response · Authors · 2025-11-29
>
> Dear Reviewer SyPJ,
> We sincerely thank you for your insightful and expert comments and the time you dedicated to reviewing our work. We thank the reviewer for appreciating the large-scale analysis (14 datasets) and the practical, training-free nature of DataProphet. We hope our response and added experiment results could address the concerns. If there’s any part we could provide a more detailed response please let us know!
>
> **On W1 (Single Model & Generality)** We acknowledge the concern regarding the reliance on InternVL3-2B. Due to the constraint on compute resources, we had to limit our experiments to this base model. To verify the generality of our findings, we applied the DataProphet pipeline using a different architecture, Qwen2.5-VL 3B. The resulting average Kendall’s τ between the predicted ranking and ground truth was 0.676. This indicates a strong correlation and suggests that the factors measuring data influence (perplexity, similarity, diversity) are transferable across different competitive MLLMs.
>
> **On W2 (Equations and Motivation)**
> Diversity: We use Silhouette coefficient + Entropy (Eq. 2) to reward both well-separated clusters (distinct topics) and balanced cluster sizes.
> Normalization: We use standard l2​ normalization for embeddings to ensure cosine similarity is scale-invariant. Perplexity is normalized by the target's perplexity to measure relative difficulty.
> Product Form: As mentioned in Section 3.2, the product form ensures that a high score requires alignment across all axes. An additive form would allow irrelevant data to score high via a single factor, which we wanted to avoid.
>
> we also evaluated an additive variant of our metric by replacing the product in Eq. (3) with a sum DONE
> | Metric (additive variant) | Avg. Kendall’s τ |
> | ------------------------- | ---------------- |
> | combined                  | 0.307      |
> | w/o question similarity                | 0.266         |
> | w/o  image similarity               | 0.241         |
> | w/o  answer similarity            | 0.298        |
> | w/o perplexity            | 0.389         |
> | w/o diversity      | 0.254        |
>
> **On W3 (Underspecified Details)** Currently, implementation details are provided in Appendix A.2. The Kendall’s τ calculation follows the standard definition as below. All runs fine-tune the InternVL3-2B base model with full-parameter supervised fine-tuning implemented in LLaMA-Factory for one epoch, using a learning rate of 1e-5. We will make these hyperparameters explicit in Appendix A.2 to improve clarity.
> Kendall’s tau is computed as follows. For each fixed target dataset t, we use the standard definition of the Kendall rank correlation coefficient. We first construct two rankings over the 14 source datasets: one based on the realized relative improvements Δ_{s→t} and one based on the corresponding DATAPROPHET scores M(s→t). Let C and D denote the numbers of concordant and discordant source pairs between these two rankings, respectively. We then compute
>
> tau_t = (C - D) / (C + D).
>
> We average tau_t over all target datasets t to obtain tau_Tgt. Symmetrically, for each source dataset s, we compare the ranking of target datasets induced by Δ_{t→s} and M(s→t) to obtain tau_s, and average over all s to get tau_Src. The final Kendall’s tau reported in Table 1 is defined as
>
> tau = (tau_Tgt + tau_Src) / 2.
>
> **On W4 (Theoretical Analysis)** we want to emphasize that we are the first work that tries to predict multimodal SFT performance gain using a training-free data-centric metric. We proved our metric’s efficacy through data selection experiments. We position this paper as an empirical study aimed at demystifying data influence through large-scale, comprehensive experimentation (14x14 matrix). While we provide heuristic motivations for our metric (Section 3.2), we believe the strong empirical correlation (τ=0.86) validates the approach. We agree that theoretical grounding is indeed a valuable direction and will be an important direction for future work.
>
> **On Minor Points & Ethics** We will incorporate the suggested references. Regarding Failures/Robustness(Question 1): DataProphet relies on the base model's perception. If the source data contains "poisoned" examples that look similar and have high perplexity but incorrect labels, DataProphet might incorrectly assign them high scores. Regarding Ethics: We acknowledge that embedding-based selection can reflect biases in the pre-trained model. This is a limitation of all embedding-based selection methods, and we will highlight this in the ethics statement. However, this is not an artifact of the proposed method itself.
>
> Dear Reviewer syPJ, thank you once again for your insightful and expert comments and the time you dedicated to reviewing our work. If our response addressed your concerns, we really hope you could consider raising your score! If it did not fully answer your questions, please let us know and we will do our best to answer these questions. Thank you.

---

### Official Review · Reviewer_KhQ8 · 2025-10-30

**Soundness:** 3
**Presentation:** 3
**Contribution:** 3
**Rating:** 6
**Confidence:** 3

**Summary:**

This paper tackles the critical, yet poorly understood, problem of data selection for training Multimodal Large Language Models (MLLMs). The authors first conduct a large-scale experiment (14 datasets, 7 tasks) to "demystify" data influence, finding that human intuition about "task similarity" is an unreliable predictor of which training datasets will improve performance on a target benchmark.Based on these counter-intuitive findings, the paper proposes DataProphet, a simple, interpretable, and training-free metric to predict a source dataset's influence on a target benchmark before any training occurs. The metric is a product of three components:Multimodal Perplexity: How "difficult" the source and target data are for a base model.Cross-dataset Similarity: Cosine similarity of text (question, answer) and visual (image) embeddings.Data Diversity: The coverage and balance of questions in the source data.The paper shows that DataProphet's predicted influence rankings are strongly correlated (86.0% Kendall's $\tau$) with the actual performance rankings obtained after SFT. Finally, it demonstrates that this metric can be used for data selection, creating training mixtures that outperform both uniform sampling and a state-of-the-art training-based selection method (ICONS).

**Strengths:**

- The paper addresses one of the most important and practical problems in modern ML: data selection.

- The fact that the DataProphet metric requires no training is its greatest strength, making it universally applicable, cheap, and fast.

- An 86.0% Kendall's $\tau$ correlation with ground-truth training outcomes is extremely high and validates the metric's effectiveness.

- The paper successfully translates the predictive metric into a practical data selection algorithm (Sec 4) and demonstrates its superiority. It outperforms uniform selection by a large margin (e.g., +6.9% on synthetic data) and, impressively, beats the computationally expensive training-based SoTA (ICONS).

- The initial analysis (Sec 2) is a valuable contribution in itself, providing concrete evidence that "intuitive task similarity" is not a reliable guide for data selection.

**Weaknesses:**

- The paper reports in Table 3 that DataProphet-guided selection (D.P.) outperforms the "Oracle" baseline on average (e.g., 71.0% vs 70.8% on real data). The "Oracle" is defined as reweighting based on the observed single-dataset improvements from Figure 2. This result is confusing and potentially counter-intuitive. How can a predictive metric beat an oracle based on the ground-truth outcomes? This implies that the optimal mixture is non-linear and that the single-dataset oracle is not the "true" oracle for a mixed-data SFT. This is a fascinating result but is not explained at all, leaving the reader to wonder if it's a profound insight or a methodological quirk. This must be clarified.

- The metric (Eq 3) is a simple product of its components. While the ablation (Table 2) shows all components are important, it's not clear why a product is the optimal combination. This feels heuristic. The paper's strength is its empirical results, but it's lighter on the theory of why this specific combination works so well.

**Questions:**

Major:
- Please explain the result in Table 3 where DataProphet (D.P.) selection outperforms the "Oracle" baseline. The Oracle is defined as using the ground-truth relative improvements from the single-dataset experiments (Fig 2). How is it possible for a predictive metric to beat the ground-truth oracle? Does this suggest that the optimal data mixture is non-linear, and that the single-dataset-training oracle is not, in fact, the true "oracle" for this task? This is a key point that needs clarification.

Minor
- How sensitive are the DataProphet rankings to the choice of the base model? The experiments use InternVL3-2B. If all components (perplexity, embeddings for similarity) were calculated using a different, popular MLLM (e.g., LLaVA-1.5), would the resulting Kendall's $\tau$ correlation still be as high?

- For the synthetic data selection (Sec 4.2), the diversity term ($Sil+H$) was removed from the metric (Eq 4). What was the reason for this? Was it computationally too expensive to calculate at the instance level, or did it simply not improve the results in this setting?

---

> ### Author Response · Authors · 2025-11-29
>
> Dear Reviewer KhQ8,
> We sincerely thank you for your detailed and insightful review of our work. We thank the reviewer for the strong endorsement, highlighting the importance of the problem and the "extremely high" correlation we achieved. We hope our response and the additional experiment results could address the concerns. If there’s any part we could provide a more detailed response please let us know!
>
> **On W1 & Major Q1 (Outperforming the Oracle)** The "Oracle" baseline in our experiments is constructed by reweighting datasets based on their individual performance gains observed in single-dataset fine-tuning (Figure 2). This approach assumes a linear additivity of data influence and prioritizes datasets that perform best in isolation. However, real-world training involves complex interactions. A greedy combination of top-performing datasets may suffer from redundancy. DataProphet (D.P.) explicitly includes a diversity term (Equation 3), which the linear Oracle lacks. We hypothesize D.P. outperforms the Oracle (+0.2% on average) because it balances quality (similarity/perplexity) with diversity, creating a more robust mixture than simply concatenating individually best-performing datasets.
>
> **On W2 (Heuristic Product Formulation)** The product formulation (Equation 3) functions as a logical "AND" gate. For effective transfer, a source dataset must be simultaneously: (1) aligned (Similarity), (2) informative (Perplexity), AND (3) diverse. If any factor is near zero (e.g., highly similar but trivial data, or diverse but irrelevant data), the transfer value should be negligible. An additive or weighted-sum formulation would allow a strong score in one dimension to compensate for a complete lack of another (e.g., totally irrelevant images getting a high score due to high perplexity), which contradicts our goal.
> | **Metric (additive variant)** |**Avg. Kendall’s τ** |
> | ------------------------- | ---------------- |
> | combined                  | 0.307      |
> | w/o question similarity                | 0.266         |
> | w/o  image similarity               | 0.241         |
> | w/o  answer similarity            | 0.298        |
> | w/o perplexity            | 0.389         |
> | w/o diversity      | 0.254        |
>
> **On Minor Q1 (Sensitivity to Base Model)** Indeed, when we train a model, the selection of the base model and training data are both highly important factors. Due to the constraint on compute resources, we had to limit our experiments to the base model InternVL3-2B. To address the concern about model dependence, we conducted additional experiments calculating Kendall’s τ using Qwen2.5-VL 3B as the base model. The average Kendall’s τ remained robust at 0.676. While slightly lower than InternVL3, this strong positive correlation still confirms that the principles of DataProphet are generalizable across different MLLM architectures and is not the artifact of a single model.
>
> **On Minor Q2 (Diversity in Synthetic Data)** We removed the diversity term for synthetic data selection (Section 4.2) purely due to computational constraints. The diversity metric (Equation 2) requires clustering the dataset to compute the Silhouette coefficient. In the synthetic setting, we select from a pool of ~1.4 million instances. Computing global clustering diversity dynamically for every candidate subset during selection is prohibitively expensive. We found that instance-level metrics (Perplexity & Similarity) provided a sufficient signal for this scale.
>
> Dear Reviewer KhQ8, thank you once again for your insightful and expert comments and the time you dedicated to reviewing our work. If our response addressed your concerns, we really hope you could consider raising your score! If it did not fully answer your questions, please let us know and we will do our best to solve these questions. Thanks.

---

### Official Review · Reviewer_aw8L · 2025-11-01

**Soundness:** 3
**Presentation:** 3
**Contribution:** 2
**Rating:** 4
**Confidence:** 2

**Summary:**

The paper explores the effectiveness of supervised data selection for training multimodal large language models (MLLMs). It introduces a novel metric, DATAPROPHET, which predicts the impact of training datasets on benchmark performance without prior training. Through empirical research on 14 diverse visual-language datasets, the authors find that intuitive task similarity does not correlate with model generalization. Instead, dataset characteristics significantly influence performance. The findings highlight the importance of careful data selection in enhancing MLLM efficacy.

**Strengths:**

Strength:

Novel Perspective on Data Selection: The article presents a new viewpoint on data selection, highlighting factors beyond mere similarity that influence model performance.

Effective Task Selection Method: The introduction of the DATAPROPHET metric has shown excellent experimental results.

Comprehensive Empirical Validation: The study includes extensive empirical validation across diverse visual-language datasets, providing robust evidence for the proposed method's effectiveness.

**Weaknesses:**

Insufficient Innovation: The proposed method lacks substantial innovation, as it appears to be merely a combination of existing metrics rather than offering a fundamentally new approach to data selection. This may limit its perceived contribution to the field.

Limited Exploration of Alternatives: The focus on the DATAPROPHET metric may overshadow other potentially valuable data selection strategies. The study doesn’t thoroughly explore how different combinations of metrics or alternative methodologies might yield even better results.

The experiment relies solely on one evaluation metric, which may lead to an incomplete assessment of the model's performance.

**Questions:**

How is the target task selected? I don't fully understand data selection. Is the ultimate goal of data selection simply to improve accuracy on the target task? How does it contribute to enhancing the model's generalization ability on new datasets?

The evaluation metric used is too singular.  We also need to consider efficiency, the sensitivity of the task data, and the sensitivity of different components of the new methods.

---

> ### Author Response · Authors · 2025-11-29
>
> Dear Reviewer,
> We sincerely thank you for your insightful and expert comments and the time you dedicated to reviewing our work. We thank the reviewer for recognizing the novelty of our perspective on data selection, the effectiveness of the DataProphet metric, and the comprehensive empirical validation across 14 datasets. We hope our response and added experiment results could address the concerns. If there’s any part we could provide a more detailed response, please let us know!
>
> **On W1 (Insufficient Innovation)** As far as we know, we are the first work that tries to predict multimodal SFT performance gain using a training-free metric, while providing a detailed multimodal generalization analysis. We agree that individual concepts used inside DataProphet (perplexity, similarity, diversity) are not new; however, determining which concepts to use and how to use them to predict the real multimodal performance gain is a very novel problem, and our work is bringing in new perspectives to an open question. Our core contribution in this paper is demystifying the correlation between training data and actual multimodal performance gain without training, breaking it down to a neat formula, which was previously unclear to the community. We further applied our findings to data selection and proved them useful. Prior work relies on "intuitive task similarity," which we empirically prove is unreliable (Figure 1). The innovation lies in identifying the specific combination of factors (Eq. 3) that reliably predicts influence (τ≈0.86) without expensive proxy training, rather than the mathematical formulation of the components themselves.
>
> **On W2 (Limited Exploration of Alternatives)** We respectfully clarify that we explicitly explored various alternative strategies at multiple experiment stages. As detailed in Section 3.3 ("Additional trials"), we tested heuristics such as question difficulty, model familiarity with images, model familiarity with questions, and answer length for designing our dataprophet metric. Empirical results showed these failed to improve correlation; for instance, adding answer length decreased τ by 0.15. Additionally, we compared multiple data selection baselines and oracle methods as in Table 3 under real data and synthetic data settings. We try to make our experiments as comprehensive as possible.  We want to emphasize that we are the first work that tries to predict multimodal SFT performance gain using a training-free metric from a data perspective, and our method proves its efficacy through our comprehensive experiments. We include the aforementioned metric-related experiment exploration results as below.
> | Metric variant         | Avg. Kendall’s τ |
> | ---------------------- |  ---------------- |
> | DATAPROPHET (product)                   | 0.860            |
> | + Image familiarity              | 0.842            |
> | + Question familiarity          | 0.695            |
> | + Answer length                      | 0.674            |
> | + Question difficulty            | 0.653            |
> | Additive counterpart  | 0.307            |
>
> **On W3 & Q2 (Evaluation Metrics and Efficiency)** We focus on Relative Performance Gain and Kendall’s τ because the primary goal of data selection is to maximize target accuracy and accurately rank data utility. The evaluation metric (accuracy) is decided by the end VQA tasks. Regarding efficiency(Q3), DataProphet is explicitly designed to be training-free and compute-light, distinguishing it from training-based baselines like ICONS that require gradient computation. Regarding sensitivity (Q3), Table 2 provides a detailed sensitivity analysis of different metric components. From Table 2, we could see that perplexity affects the most – removing perplexity will make the kendall’s τ drop from 0.86 to 0.487, and that image similarity is the second most important factor – removing which will make the kendall’s τ drop from 0.86 to 0.625.
>
> **On Q1 (Target Task Selection and Generalization)** Multimodal target tasks were selected to maximize diversity, covering 7 distinct families (OCR, Chart, Spatial, Counting, General VQA, Document, Map) and 14 most popularly used specific tasks to simulate realistic generalization scenarios. The goal of data selection is to prove that using our proposed DataProphet, we could select data easily without any training, while being able to improve performance on these target distributions. It contributes to generalization by identifying source data that effectively transfers capabilities to the target domain, even when the data types appear superficially different (as seen in Figure 2 where OCR data improves Spatial tasks).
>
> Dear Reviewer aw8L, thank you once again for your comments and the time you dedicated to reviewing our work. If our response addressed your concerns, we really hope you could consider raising your score! If it did not fully answer your questions, please let us know and we will do our best to resolve the questions. Thanks.

---

### Meta-Review · Area_Chair_wNrA · 2026-01-06

**Summary:**

This paper investigates data selection for supervised fine-tuning (SFT) of Multimodal Large Language Models (MLLMs).
Through an empirical analysis of 14 datasets across 7 task families, the authors demonstrate that "intuitive task similarity" (e.g., text-rich tasks helping other text-rich tasks) is an unreliable predictor of performance gains.
Authors propose DATAPROPHET, a training-free metric that combines multimodal perplexity, cross-dataset similarity (text and image embeddings), and data diversity to predict a dataset's influence on a target benchmark. The metric shows a high Kendall's correlation (86.0%) with actual post-training performance rankings.
 Experimental results indicate that DATAPROPHET-based selection outperforms uniform sampling, state-of-the-art training-based methods (ICONS), and even an experimental performance-based oracle.

Thus, I recommend accepting this paper.

**Reviewer Concerns:**

1 In rebuttal, the authors have largely addressed reviewers' concerns by providing more results such as adding Qwen2.5-VL 3B, and giving more clarifications.

2 Still outstanding:
Reviewer LeY5 raised concerns regarding "task-intrinsic improvability," suggesting that some benchmarks might show higher gains simply because they are easier to improve.

**Reviewer Scores:**

The initial ratings were predominantly cautious, with three reviewers at ''4'' (marginally below threshold) and one at ''6'' (marginally above threshold).

After rebuttal, Reviewer aw8L might remain concerned about the "combination of existing metrics"; and Reviewer LeY5 remained critical of the underlying analytical framework.
The other two reviewers (KhQ8 and SyPJ) may increase their ratings.

---

### Decision · Program_Chairs · 2026-01-26

Accept (Poster)